# Disentangling the Genetic Landscape of Peripartum Depression: A Multi-Polygenic Machine Learning Approach on an Italian Sample

**DOI:** 10.3390/genes15121517

**Published:** 2024-11-26

**Authors:** Yasmin A. Harrington, Lidia Fortaner-Uyà, Marco Paolini, Sara Poletti, Cristina Lorenzi, Sara Spadini, Elisa M. T. Melloni, Elena Agnoletto, Raffaella Zanardi, Cristina Colombo, Francesco Benedetti

**Affiliations:** 1Vita-Salute San Raffaele University, 20132 Milan, Italy; y.harrington@studenti.unisr.it (Y.A.H.); l.fortaneruya@studenti.unisr.it (L.F.-U.); poletti.sara@hsr.it (S.P.); zanardi.raffaella@hsr.it (R.Z.); colombo.cristina@hsr.it (C.C.); 2Psychiatry & Clinical Psychobiology, Division of Neuroscience, IRCCS San Raffaele Hospital, 20132 Milan, Italy; paolini.marco@hsr.it (M.P.); lorenzi.cristina@hsr.it (C.L.); spadini.sara@hsr.it (S.S.); melloni.elisa@hsr.it (E.M.T.M.); agnoletto.elena@hsr.it (E.A.); 3Mood Disorders Unit, IRCCS San Raffaele Hospital, 20132 Milan, Italy

**Keywords:** peripartum depression, polygenic risk scores, partial least squares regression, major depressive disorder, bipolar disorder

## Abstract

Background: The genetic determinants of peripartum depression (PPD) are not fully understood. Using a multi-polygenic score approach, we characterized the relationship between genome-wide information and the history of PPD in patients with mood disorders, with the hypothesis that multiple polygenic risk scores (PRSs) could potentially influence the development of PPD. Methods: We calculated 341 PRSs for 178 parous mood disorder inpatients affected by major depressive disorder (MDD) or bipolar disorder (BD) with (*n* = 62) and without (*n* = 116) a history of PPD. We used partial least squares regression in a novel machine learning pipeline to rank PRSs based on their contribution to the prediction of PPD, in the whole sample and separately in the two diagnostic groups. Results: The PLS linear regression in the whole sample defined a model explaining 27.12% of the variance in the presence of PPD history, 56.73% of variance among MDD, and 42.96% of variance in BD. Our findings highlight that multiple genetic factors related to circadian rhythms, inflammation, and psychiatric diagnoses are top contributors to the prediction of PPD. Specifically, in MDD, the top contributing PRS was monocyte count, while in BD, it was chronotype, with PRSs for inflammation and psychiatric diagnoses significantly contributing to both groups. Conclusions: These results confirm previous literature about the immune system dysregulation in postpartum mood disorders, and shed light on which genetic factors are involved in the pathophysiology of PPD.

## 1. Introduction

The periods during pregnancy and shortly after childbirth are particularly vulnerable time points for women to experience disturbances in mood. Specifically, peripartum depression (PPD) is defined as a depressive episode that occurs during pregnancy or the period after delivery and is characterized by feelings of sadness, guilt, and lack of motivation [1]. The global prevalence of PPD is roughly 17% [2], yet in reality, the number of women suffering from PPD is most likely much higher, as 50% of cases are believed to be underreported or untreated [3]. Currently, PPD is classified as a subtype of major depressive disorder (MDD) in the latest DSM edition [4]; however, PPD can also exist in the context of a bipolar disorder (BD) diagnosis, with around a 3% prevalence in postpartum women [5]. Some believe PPD to be a distinct clinical entity with unique characteristics and risk factors. Most research focused on PPD has been in comparison to healthy controls, with only a handful of studies exploring the differences between PPD, MDD, and BD with mixed findings [6,7].

The etiology of PPD is complex as it is believed to be multifactorial involving psychological, social, and biological elements. Ongoing research has been conducted in multiple avenues of potential risk factors. Researchers have linked prior episodes of psychiatric illness and comorbid psychiatric disorders as a strong predictor of developing a PPD episode. Indeed, Bloch and colleagues [8] found that a prior history of MDD was the strongest predictor for developing a PPD episode and that PPD had a 30% incidence rate in women with such prior history. In another longitudinal study following women with and without prior history of depression or premenstrual dysphoric disorder, they found women with a history of both diagnoses were twice as likely to develop PPD [9]. Similarly, women with a history of BD also have an increased risk of mood disturbances during the perinatal time period [10]. Furthermore, researchers have demonstrated that the majority of women referred to clinical treatment for PPD were indeed bipolar [11].

Moreover, the unique biology of pregnancy itself may result in increased risk of PPD in vulnerable individuals. Specifically, hormones like estrogen, progesterone, and testosterone, which reach abnormally high levels during pregnancy and quickly drop shortly after birth, could serve as a trigger for alterations in mood. Some argue that there is a subgroup of women that are particularly sensitive to fluctuations in hormone levels and thus may be more likely to develop depressive symptomatology during these periods of extreme hormone level alterations [12]. Following this line of reasoning, one study simulated the hormonal changes seen during the perinatal period and found that only women with a prior history of PPD developed depressive symptoms [13].

In addition, inflammatory markers and immune related conditions are altered during pregnancy. Elevated levels of inflammatory cytokines and decreased T cell activation during pregnancy and postpartum periods are linked to an increased risk of peripartum depression [14,15,16], and indeed some authors hypothesized that the immune system of PPD women may be specifically less adaptable to stressors [14].

Furthermore, disrupted sleep patterns and misalignment of circadian rhythms, which are common during pregnancy and after childbirth, can contribute to the development of peripartum depression. Studies have linked sleep quality, sleep disturbances, and alterations in circadian rhythms to PPD [17,18,19]. Specifically, a study linked sleep timing to the development of PPD symptoms where women with a later sleep phase had higher rates of depressive symptomatology [20].

In recent years, there has been an ever-growing focus on the genetics of psychiatric disorders. Like many psychiatric conditions, having a familial history of mental illness is a strong risk factor for PPD, suggesting a possible genetic link [21]. Investigators have focused on twin and sibling studies as well as on specific genes and loci [22]. Twin studies found the heritability of PPD to be around 50%, generally higher than what found in MDD [23]. The same study found that 14% of the genetic variance of PPD was unique to this condition, suggesting that PPD may be in part distinct from MDD. Besides heritability studies, researchers have also tried focusing on specific genes. Several candidate gene studies have identified associations between single nucleotide polymorphisms (SNPs) and the risk of PPD [24]. However, given the complexity of PPD, it is unlikely that its heritability can be fully explained by individual SNPs. Supporting this, a recent meta-analysis concluded that no single candidate genes or gene sets reliably predict depression phenotypes [25].

An alternative approach that holds great promise relies on the use of polygenic risk scores (PRSs), which aggregate the effects of numerous genetic variants identified through genome-wide association studies (GWASs). PRSs employ weighted measures of multiple SNPs associated with a given phenotype to explain a larger proportion of genetic variance. Recent evidence suggests that polygenic risk prediction captures phenotypic variance more effectively than SNP-based heritability alone [26]. Moreover, PRSs might very well prove to be potential tools in clinical practice, offering utility in screening for mental health disorders, improving diagnostic accuracy, guiding clinical decisions, and predicting treatment response and adverse health outcomes [27]. Despite the growing utility of PRSs in psychiatric research, a critical methodological challenge remains unaddressed: identifying which PRSs are the strongest predictors for a specific phenotype. In addition, many psychiatric disorders and medical conditions share genetic underpinnings, thus leading to high multicollinearity among PRSs, which further complicates the selection of the most informative genetic predictors.

While recent research has tried to investigate the relationship between single PRSs and PPD [28,29], here we adopt a more powerful approach by leveraging the joint predictive power of multiple PRSs. Specifically, we utilize a multi-polygenic score framework that integrates genetic correlations across numerous traits within a single regression model predicting PPD history within a sample of parous female patients diagnosed with mood disorders. To achieve this, we employed a partial least squares (PLS) regression method within a novel machine learning pipeline, an approach which is particularly effective for addressing multicollinearity among predictors and for managing datasets where the number of independent variables exceeds the number of samples [30].

## 2. Materials and Methods

### 2.1. Participants 

Our sample of female patients with and without PPD history was retrieved from a larger cohort of 435 individuals with MDD and BD, who were consecutively enrolled in an ongoing prospective study conducted at IRCCS San Raffaele Hospital, Milan (Italy). These patients were referred consecutively to the Mood Disorder Unit by their general practitioners or outpatient psychiatrists for psychopathological conditions requiring hospitalization. Mood disorder diagnoses were made following a procedure including evaluation and diagnosis based on a psychiatric interview (DSM-IV criteria) by the team responsible for the admission to the ward. The diagnosis was confirmed by a senior psychiatrist that specializes in mood disorders using a best-estimate method. This process involved interviews with patients, their family members, and prior healthcare providers, along with a review of available medical records [31]. At the time of admission, ongoing treatments were administered based on clinical necessity.

We specifically analyzed 178 parous women aged between 28 and 70 diagnosed with MDD or BD, 62 of whom had a history of PPD, while 116 had no such history. In this study, a peripartum episode was defined as a major depressive episode occurring during pregnancy or within one year post-partum, according to guidelines from the World Health Organization, the Centers for Disease Control and Prevention, and the American College of Obstetricians and Gynecologists [32,33,34]. Peripartum status was documented using patients’ medical records based on a clinical interview performed by a psychiatrist to gather patient psychiatric history.

To be included in the current study, participants had to have had at least one pregnancy [35] and availability of blood sampling data. Exclusion criteria included age under 18 or over 70, the presence of additional Axis I psychiatric disorders, intellectual disability, major medical or neurological conditions, and a history of substance or alcohol abuse within the past six months. These criteria were verified through patient medical charts. Written informed consent was obtained from all participants after a full explanation of the study. The study protocol adhered to the Declaration of Helsinki and was approved by the local Ethics Committee from the IRCSS San Raffaele Hospital. Since our sample was retrospective, our sample size was determined by screening the patients that had been recruited in an ongoing larger study of mood disorders (*n* = 435) using our inclusion and exclusion criteria. We believe our sample is representative of the population of individuals with a history of PPD in patients with mood disorder diagnoses, as previous literature has observed a 40–50% prevalence of PPD among major patients affected by major depressive and bipolar disorder [36].

### 2.2. Polygenic Risk Score Calculation 

Genotyping, quality control (QC), and imputation of genetic data were conducted within the above-mentioned larger cohort of 435 patients diagnosed with mood disorders, including 179 individuals with MDD and 256 with bipolar disorder. The current sample of 178 patients is a subset of this cohort. Blood samples were genotyped using the Infinium PsychArray 24 BeadChip (Illumina, Inc., San Diego, CA, USA). This cost-efficient and high-density micro-array was developed in collaboration with the Psychiatric Genomics Consortium and includes about 548,000 tag SNPs, with additional 50,000 markers previously associated with psychiatric disorders (https://www.illumina.com/products/by-type/microarray-kits/infinium-psycharray.html) (accessed on 12 July 2024).

QC procedures were carried out using PLINK1.9 [37]. Participants were excluded if they exhibited genotyped sex mismatches as compared to the phenotype, a genotype rate below 95%, or outlying autosomal heterozygosity (Fhet > ±0.2). Genetic markers were filtered out if they had a minor allele frequency (MAF) below 1%, a call rate lower than 95%, or deviated from the Hardy–Weinberg equilibrium at *p* < 10^−6^. Relatedness of participants was assessed by excluding individuals with a degree of recent shared ancestry (identity by descent) greater than 0.1875, a threshold between third- and second-degree relatives [38]. European ancestry of the sample was confirmed through principal component analysis (PCA), and individuals whose genotype distribution deviated by more than five standard deviations from the mean of the first two components were removed.

Following QC, genotype imputation and PRS calculation were performed using the Michigan Imputation Server [39]. The imputation process employed the 1000 Genomes Project V5 as the reference panel. The Eagle V2.3 algorithm was used for genotype phasing, and Minimac3 was used to impute the phased haplotypes. After imputation, 4870 PRSs were calculated by exploiting pgsc_calc (https://github.com/PGScatalog/pgsc_calc) (accessed on 12 July 2024), an automated workflow that estimates PRS for all scoring files publicly available in the Polygenic Score Catalog [39].

Among the 4870 PRSs calculated, we selected a total of 341 PRSs related to our traits of interest: 67 PRSs for psychiatric symptoms and disorders, 89 PRSs for hormones and pregnancy-related conditions, 131 PRSs for immune-inflammatory markers, and 54 PRSs for circadian rhythms and sleep-related problems. For reference, a full list of all the PRSs included in our study and their corresponding source articles including the information regarding the original GWASs and their Polygenic Score Catalog codes is provided in Appendix A.

### 2.3. Statistical Analysis

To evaluate the effect of PRSs on the history of PPD, we applied partial least squares (PLS) regression, a data mining technique that models relationships between observed variables and latent variables, in the context of an innovative machine learning pipeline. PLS defines a linear regression model by projecting both the predictor and response variables into a new latent space, enabling feature reduction while maximizing the covariance between them [30,40,41]. We selected PLS regression given its ability to handle high-dimensional predictors, such as the 431 PRSs included in this analysis, which served as predictors. Furthermore, PLS regression is able to condense the information from all the included predictors into latent variables, without removing any of the predictors as other penalized regressions do. The diagnosis of PPD versus non-PPD was entered as the dependent variable, while mood disorder diagnosis (MDD vs. BD) was included as a covariate only in the model including the entire sample.

PLS regression was employed to rank the PRSs based on their contributions to PPD prediction. The Nonlinear Iterative Partial Least Squares (NIPALS) algorithm was used to optimize the model by selecting the appropriate number of PLS components to extract (A) via a k-fold cross-validation with 7 folds. This process allowed us to calculate key metrics such as R^2^X (an A-dimensional vector representing the variance in the predictor matrix explained by each PLS component), and R^2^Y (representing the variance in the response variables explained by each PLS component). Additionally, the model generated predictive weights (w) and variable importance in projection (VIP) scores for each PRS, with VIP values greater than 1 indicating significant contributions to the model, as suggested by previous literature [42,43,44]. These metrics quantified the contribution of each PRS to explaining variance in the clinical outcome (Y) and determined the direction of the effect.

By ranking PRSs based on their VIP scores, we were able to identify the most important genetic contributors to PPD. This method effectively captures the influence of multiple independent biological variables and has been validated in previous randomized controlled trials (RCTs) involving mood disorder patients with a combination of biological and clinical predictors [45]. To ensure robustness and prevent overfitting, the model was validated using a cross-validation approach. In addition, this pipeline was applied separately in both MDD and BD diagnostic groups, in order to assess how genetic risk factors for PPD may differ between these mood disorders. Finally, sensitivity and specificity values for the first latent variable extracted from each PLS regression model were estimated to calculate the area under the receiver operator curve (AUC) of the model.

## 3. Results

Demographics of the sample can be found in Table 1. Age was the only factor significantly different between the two groups. PPD women may be significantly younger as they had a depressive episode during their reproductive window, while women who do not have a PPD history may be older as they may not have developed depressive symptomatology until later in life.

### 3.1. Whole Sample

The PLS linear regression in the whole sample (*n* = 178, 62 PPD and 116 no PPD), using the 341 PRSs and mood disorder diagnosis as factors, defined a model with one extracted component explaining 27.12% of the variance in the presence of PPD history (coefficient = 0.1997, R^2^X = 0.0347; R^2^Y = 0.2712). In addition, 106 PRSs exceeded VIP = 1 and were selected as significant in contributing to explaining variance. The top five contributing PRSs were PGS002390 Chronotype (morning) (w = −0.1671, VIP = 3.0945), PGS002439 Chronotype (morning) (w = −0.1553, VIP = 2.8766), PGS002609 Monocyte count (w = 0.1510, VIP = 2.7971), PGS002560 Monocyte count (w = 0.1496, VIP = 2.7710); PGS001163 Monocyte count (w = 0.1432. VIP = 2.6520). The top 20 contributing PRSs can be found in Figure 1A and the full results showing the ranking and weight of the 106 PRSs with a VIP greater than 1, and thus are significant contributors to the PPD prediction model, can be found in Appendix A. The first latent variable of the whole sample PLS model obtained an AUC of 0.823 (Figure 2A).

### 3.2. MDD Sample

The PLS linear regression in the MDD sample (*n* = 72, 25 PPD and 47 no PPD), using the 341 PRSs as factors, defined a model with one extracted component explaining 56.73% of the variance in the presence of PPD history (coefficient = 0.2310, R^2^X = 0.0425; R^2^Y = 0.5673). In addition, 107 PRSs exceeded VIP = 1 and were selected as significant in contributing to explaining variance. The top five contributing PRSs were PGS001163 Monocyte count (w = 0.1419, VIP = 2.6209), PGS002560 Monocyte count (w = 0.1352, VIP = 2.4973), PGS000675 CRP (−0.1351, VIP = 2.4939), PGS002868 CRP (w = −0.1336, VIP = 2.4670); PGS002609 Monocyte count (w = 0.13357, VIP = 2.4665). The top 20 contributing PRSs can be found in Figure 1B and the full results showing the ranking and weight of the 107 PRSs with a VIP greater than 1, and thus are significant contributors to the PPD prediction model, can be found in Appendix A. The first latent variable of the MDD sample PLS model obtained an AUC of 0.948 (Figure 2B).

### 3.3. BD Sample

The PLS linear regression in the BD sample (*n* = 106, 37 PPD and 69 no PPD), using the 341 PRSs as factors, defined a model with one extracted component explaining 43.96% of the variance in the presence of PPD history (coefficient = 0.2686, R^2^X = 0.0335; R^2^Y = 0.4396). In addition, 98 PRSs exceeded VIP = 1 and were selected as significant in contributing to explain variance. The top five contributing PRSs were PGS002439 Chronotype (morning) (w = −0.1931, VIP = 3.5649), PGS002390 Chronotype (morning) (w = −0.1885, VIP = 3.4801), PGS002787 Type 1 bipolar disorder (w = −0.1769, VIP = 3.2667), PGS000756 Narcolepsy (w = −0.1765, VIP = 3.2598) and PGS002786 bipolar disorder (w = −0.1626, VIP = 3.003). The top 20 contributing PRSs can be found in Figure 1C and the full results showing the ranking and weight of the 98 PRSs with a VIP greater than 1, and thus are significant contributors to the PPD prediction model, can be found in Appendix A. The first latent variable of the BD sample PLS model obtained an AUC of 0.904 (Figure 2C).

### 3.4. Comparison Between MDD and BD Models

Different distributions of the PRSs were included in the MDD (Table 2) and BD models (Table 3). For the MDD sample, 29.8% of the psychiatric PRSs were included in the model, with most positively predicting PPD. In addition, 27% of hormone and pregnancy PRSs were included in the model, with slightly more having a negative association with PPD. For immune PRSs, 35.9% were included in the model with most having a negative association with PPD. Lastly, 29.6% PRSs for sleep and circadian rhythms were included in the model with slightly more being negatively associated with PPD. Immune-related PRSs had the highest contribution to the model for MDD PPD, which was driven mostly by PRSs for monocytes in the positive direction and C-reactive protein (CRP) in the negative direction.

For the BD sample, 19.4% of the psychiatric PRSs were included in the model with most having a negative association with PPD. For hormone and pregnancy PRSs, 37.1% of PRSs were included in the model, with slightly more having a positive association with PPD. In addition, 21.4% of immune PRSs were included in the model, with most having a positive association with PPD. Lastly, for sleep and circadian PRSs, 44.4% were included in the model with more having a negative association with PPD. Sleep and circadian PRSs had the highest contribution to the BD PPD model, with morning chronotype driving the association in the negative direction and insomnia, sleep duration, and trouble falling asleep driving the positive direction.

Overall, psychiatric PRSs had a more positive effect in MDD and a more negative effect in BD, while the opposite was found in immune and inflammation PRSs with a more negative effect in MDD and a positive one in BD. The differing effects of immune and inflammation related PRSs seemed to be driven by the opposite weights for CRP PRSs in the two diagnostic groups. Lastly, in the BD model, there was a higher percentage of hormone and pregnancy-related PRSs suggesting the genetic components related to these traits have a higher impact in BD PPD than MDD PPD.

## 4. Discussion

Our study aimed to investigate the genetic underpinnings of PPD by leveraging a novel methodological approach that integrates multiple PRSs into a predictive model. Previous research examined the role of genetic risk factors in PPD by employing regression models based on individual PRSs [28,29], limiting their capacity to account for the complex genetic architecture influencing this disorder. In contrast, our approach utilizes a multi-polygenic score framework, which has been applied for the first time to the prediction of PPD. This methodology is particularly relevant for complex psychiatric disorders like PPD, which are unlikely to be driven by single genetic variants but rather by the cumulative effect of numerous variants across various biological pathways [25]. Additionally, we exploited PLS regression within a machine learning framework that optimizes the prediction model by managing the high-dimensionality and multicollinearity inherent in genetic data. One of the advantages of such a methodology is that it allowed us to assess the joint contribution of multiple PRSs to the risk of PPD, thereby addressing the challenge of genetic correlation that arises from pleiotropy or shared biological mechanisms. Indeed, many psychiatric and medical conditions share overlapping genetic risk factors, leading to significant correlations among PRSs [46]. Traditional regression models often struggle to disentangle these effects, potentially missing key genetic information. By employing PLS regression, which reduces dimensionality while maximizing covariance between predictors and the outcome, our method effectively handled multicollinearity, allowing us to identify the most relevant genetic predictors for PPD. This is a critical advantage, as it enhances the interpretability of our results and provides clearer insights into the genetic architecture of PPD.

Our study investigated the genetic underpinnings of PPD, showing that only partially overlapping PRSs are associated with PPD in MDD and BD patients. Indeed, variance explained by the PRSs dropped from 57% and 44% when analyzing the MDD and BD samples separately to 27% when analyzing all participants together. This suggests that the contribution of the same PRSs to the development of PPD in the two conditions may be different, thus increasing the error when these factors are modelled together. This perspective is in agreement with recent findings showing that the genetic landscapes for the two disorders have different effects in influencing the gene–environment interaction preceding an illness episode [47], epigenetically active sites in immune cells [48], and several circulating biomarkers, including immune-inflammatory markers [49]. This observation suggests that the genetic contribution to PPD could involve different factors in the two disorders.

Accordingly, in the MDD subsample, inflammation-related PRSs were the most predictive for PPD, with a smaller but still significant impact in the BD sample. Despite a growing body of evidence, the relationship between immune-inflammatory alterations and mood disorders is far from being completely understood. MDD is thought to be characterized by a chronic inflammatory status, which may foster episode recurrences [50]. Many of the PRSs found to be associated with PPD in MDD patients in our study were related to monocytes. Monocytes are white blood cells that are a part of the innate immune system and are thought to play a role in chronic inflammation [50]. Furthermore, monocytes are able to cross the blood–brain barrier and therefore may alter the inflammatory state of the central nervous system. Previous studies have found elevated levels of monocytes as well as genetic evidence of glucocorticoid resistance in monocyte cells in cases of MDD [51]. MDD monocytes have showed an overexpression of gene clusters related to inflammation, apoptosis, and premature aging [52]. In postpartum psychosis, a study showed increased monocyte levels and increased glucocorticoid resistance in monocytes in affected individuals compared to healthy mothers [53]. In regard to PPD, there is limited research exploring the direct role of monocytes in the disorder’s pathology. However, in one longitudinal study, researchers found higher levels of prenatal monocyte activation in women with postpartum depressive symptoms compared to healthy mothers [54]. Monocytes can release proinflammatory cytokines, which have themselves been linked to the presence of PPD symptomatology. Osborne and colleagues [15] found elevated levels of chemokine (C-C motif) ligand 3 (CCL3), which is a potent proinflammatory chemokine that is released from and also attracts monocytes in depressed pregnant women compared to healthy women. Moreover, the role of monocyte dysfunction has been previously associated with environmental effects, such as exposure to childhood trauma or infection [52,55]. Our findings confirm a key role for monocytes in PPD, by associating for the first time genetic factors affecting monocyte counts with the diagnosis.

C-reactive protein (CRP) PRSs were also among the top contributing factors in all our models, mostly exhibiting an inverse association with PPD risk. CRP is a key inflammatory marker and its association with mood episodes has been repeatedly demonstrated [56,57] as well as with PPD symptomatology in previous studies [58]. However, findings are mixed as one study found that CRP levels were not associated with PPD diagnosis and treatment response [59], suggesting that CRP levels may not be as important for the etiology of PPD as it is for MDD. In addition, PRSs for CRP may contain gene variants that are protective against MDD, thus suggesting that genotype and circulating levels of CRP could be independently associated with depression [60]. Moreover, PRS for CRP and CRP plasma levels were shown to have opposite effects on antidepressant response, with CRP PRS being positively associated with antidepressant response and circulating levels of CRP showing a negative association with response [61], thus confirming that many factors affecting this biomarker, independently from its PRS, might have a detrimental effect on mood disorders. Yet, to our knowledge, no study has directly compared CRP levels between PPD and MDD cases.

PRSs for other proinflammatory markers were also found to be associated with PPD in our study, including tumor necrosis factor receptors and galectin 3. Interestingly, a different immunological profile appeared to characterize the two diagnoses, as CC-chemokine 20 and interleukin 27 (IL27) as well as autoimmune diseases PRSs were negatively associated with PPD in the BD model but did not significantly contribute to the MDD model. This confirms that cytokines and chemokines have different associations with the diagnosis in the two disorders [62,63], extending this perspective to PPD.

A different genetic profile appeared to characterize the BD sample, where the highest ranking PRSs were those related to chronotype and sleep disorders. This appears to be in line with abundant evidence linking the incidence and prognosis of BD to chronotype and sleep patterns. Chronotype refers to the individual variability in circadian rhythms that are driven by the internal biological clock [64]. A later evening chronotype has been thought to be linked with depressive symptomatology possibly through its association with sleep debt accumulation, which in turn can cause misalignment of the biological clock [65]. Studies have demonstrated that later chronotype and alterations in circadian rhythms predict more frequent and more severe depressive episodes and less frequent manic episodes in BD patients [66,67]. Furthermore, a study noted that MDD patients and healthy controls with an evening chronotype tend to have enhanced characteristics of bipolarity compared to individuals with an earlier chronotype [65,68]. Moreover, one study comparing circadian disturbances between individuals with BD and MDD found that BD was associated with more desynchronization of the biological clock compared to MDD, suggesting that circadian rhythm alterations could specifically characterize BD subjects [69]. In our study, PRS for morning chronotype showed a negative association with PPD history, meaning that individuals with the genetic predisposition for an evening chronotype could be at a higher risk for a PPD episode. Prior studies have linked evening chronotypes to more severe perinatal depressive symptomatology as well as other pregnancy-related issues known to affect the development of PPD such as gestational diabetes and cesarean section [70]. It is possible that in PPD, the genetic predisposition for a later chronotype could affect sleep schedules during the peripartum periods, contributing to the development of psychopathology. In this regard, melatonin signaling alterations could be key contributors to altering chronotype shifting in peripartum women [71]. Indeed, increased nighttime light and decreased daylight exposure may alter melatonin levels and patterns of secretion [72,73,74]. A negative association between sleep disorders and PPD seems counterintuitive as previous research has shown that sleep problems during the peripartum period may increase the risk of PPD [75]. However, others have suggested that only sleep quality and not sleep disorders are linked to PPD [76]. Furthermore, the PRS for trouble falling asleep was positively associated with PPD, further confirming that genetic predispositions to symptomatology of delayed sleep are associated with PPD.

PRSs related to pregnancy conditions and hormones were also found to be associated with previous history of PPD in our sample. For instance, placenta growth factor (PGF) was positively associated with PPD history. Although its direct relationship with mood disorders both during the peripartum period and beyond has not been explored, dysfunction in placental health has been associated with an increased risk of PPD [77]. In addition, sex hormone binding globulin (SHBG) was also found to affect PPD status. The production of this protein, which has a key role in transporting androgens and estrogens and regulating the amount of free hormones available for tissues to utilize [78], physiologically increases up to ten-fold during pregnancy [79]. Interestingly, one Mendelian randomization study found a significant positive correlation between SHBG and risk of depression only in females, underlying its possible specific role in estrogen-related mood episodes [80]. Finally, higher PRSs for age at menopause were associated with PPD in the BD model, which could reflect a complex interplay between genetic factors influencing reproductive hormones and mood regulation. Both menopause timing and mood disorders are influenced by hormonal fluctuations [81]. The genes that affect menopause timing might also influence how sensitive a woman is to changes in hormone levels during the peripartum period. The genetic predisposition for a later age at menopause could imply that a woman has genes promoting longer reproductive function and, therefore, a more prolonged exposure to more cycles of fluctuations of steroid hormones known to be involved in depression [82].

Concerning psychiatric genetic risk, PRSs for MDD and BD were significant contributing factors for PPD. Surprisingly, the PRSs for MDD were negatively associated with PPD in all models. Previous studies employing healthy controls found mostly positive associations for this genetic trait [29,83], and yet comparisons between cases of PPD and other mood disorders are mixed [84,85]. Similarly, BD PRSs were negatively associated with PPD in the BD sample. On the other hand, in the MDD sample, BD PRSs were positively associated with PPD. One other study found similar results suggesting that PPD cases share more genetic overlap with BD than MDD cases do [28], supporting the long-held clinical view linking postpartum episodes to bipolar vulnerability [86]. In our sample, all patients had a diagnosis of either MDD or BD, thus sharing a common genetic landscape for mood disorders. Therefore, in light of our findings revealing the role of genetic factors related to hormones, inflammation, and chronotype contributing to PPD, our results suggest that the contribution of genes associated with these other factors associated with PPD could be independent from the effect of genes related to BD and MDD diagnosis.

Lastly, when observing results in the whole sample, while some association between PRSs and PPD seemed to be driven by either MDD (inflammation) or BD (circadian rhythms and sleep disorders) samples, the contribution of other PRSs seemed to be independent of diagnosis, such as obesity and duration of sleep. Obesity PRSs were negatively associated with PPD, suggesting that having lower scores for obesity is protective for developing PPD in the context of mood disorders. These effects could be due to the protective physiological and hormonal changes that occur during pregnancy, including metabolic adaptations that counterbalance the effects of obesity-related insulin resistance and an anti-inflammatory immune shift. These changes may temporarily mitigate the risk factors associated with obesity that typically increase the risk of depression [87], thereby lowering the risk of peripartum depression in individuals with a genetic predisposition to obesity. Interestingly, self-reported sleep duration had a positive association with PPD in the whole sample, suggesting that women who are genetically predisposed to sleep longer have an increased risk of PPD. Both short and long sleep durations have been shown to be associated with psychiatric disorders [88]. One study, examining the effects of aggregate SNPs contained in the PRS for sleep duration, found positive associations between the genetics scores of sleep duration and psychiatric illness [89]. In addition, it is possible that women who are predisposed to sleeping longer may have a harder time adjusting to shorter sleep times associated with the perinatal period.

However, our study should be viewed in light of some limitations. First, this study was conducted in a sample of Italian women from one center, possibly limiting the generalizability of the results. Furthermore, despite the specialized selected sample, the sample size could be considered small for a genetics study. Moreover, our sample was derived from a larger cohort that included male subjects, thus potentially biasing the imputation procedure. In addition, in our models, there are multiple PRSs for the same phenotype that were derived from different source articles using various calculation and imputation methods, which may skew the predictive power of the analysis. However, we attempted to mitigate this issue by using a PLS regression to combat multicollinearity. Lastly, without a healthy control comparison, we do not know how well these PRSs predict PPD diagnosis outside of mood disorder patients and in the general population. Therefore, larger samples from multiple centers including only one PRS per trait using parous female patients for imputation procedures and a matched healthy control group are needed to confirm these results.

## 5. Conclusions

PPD is complex and there are numerous avenues of potential interest related to its biological etiology and what separates it from MDD and BD. Therefore, in this study, we used multiple PRSs from four different categories including psychiatric symptoms and disorders, hormones and pregnancy-related conditions, immune-inflammatory markers, and circadian rhythms and sleep-related problems in a machine learning approach to predict positive history of PPD. From our results, it is evident that the genetics related to inflammation are important for the development of PPD in the context of MDD and in BD, genetics for sleep alterations and circadian rhythms are the most impactful. Overall, our study introduces an innovative and powerful methodology for investigating the genetic basis of PPD. By employing a multi-polygenic score approach and leveraging advanced machine learning techniques, we provided an accurate prediction model disentangling PPD risk among mood disorders. This not only advances the field methodologically but also paves the way for future studies to further explore the genetic mechanisms underlying PPD and other psychiatric conditions.

## Figures and Tables

**Figure 1 genes-15-01517-f001:**
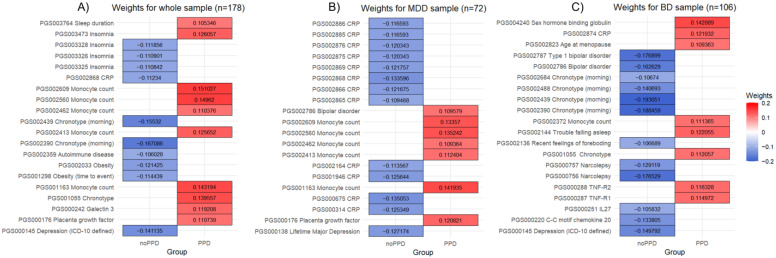
Mean weights of the top-ranked variables according to their variable importance in projection (VIP) from the partial least squares regression model on (**A**) the whole sample (*n* = 178), (**B**) the MDD sample (*n* = 72), and (**C**) the BD sample (*n* = 106). Abbreviations: CRP, C-reactive protein; TNF-R, tumor necrosis factor—receptor; IL, Interleukin.

**Figure 2 genes-15-01517-f002:**
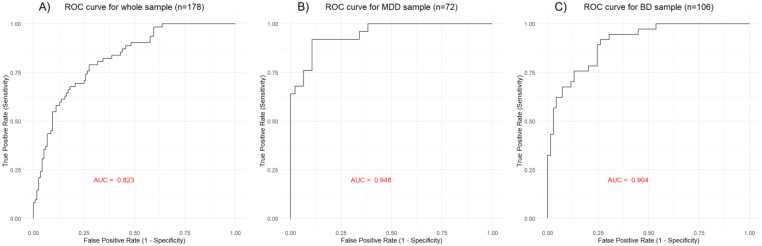
Receiver operator curve (ROC) of the first latent variable from the partial least squares regression on (**A**) the whole sample (*n* = 178), (**B**) the MDD sample (*n* = 72), and (**C**) the BD sample (*n* = 106). Abbreviations: AUC, area under the receiver operator curve; ROC, receiver operator curve.

**Table 1 genes-15-01517-t001:** Demographic table and statistics of the whole sample.

Variables	PPD (*n* = 62)	No PPD (*n* = 116)	t/x^2^	Cohen’s D/Cramer’s V	*p*-Value
Age (years)	48.18 ± 9.40	53.85 ± 8.89	−3.976	0.620	<0.001
Education (years)	12.05 ± 4.19	10.91 ± 3.79	1.846	0.285	0.067
BMI	26.38 ± 5.51	25.80 ± 4.80	0.683	0.112	0.496
Smoking Status	37/79	20/42	1.938	0.104	0.585
Alcohol Status	4/58	3/113	4.410	0.157	0.353
Number of pregnancies	2.07 ± 1.05	1.95 ± 0.99	0.732	0.118	0.465
Sex of children (male/female)	50/52	100/70	2.477	0.118	0.116
Menopause (yes/no)	21/41	67/49	9.223	−0.228	0.002
Current hormone replacement therapy (yes/no)	3/59	9/107	0.548	−0.055	0.459
Number of PPD episodes	1.21 ± 0.52	NA	NA	NA	NA
Duration of illness (years)	19.31 ± 11.65	18.45 ± 11.65	0.468	0.074	0.640
Number of mood episodes	9.86 ± 12.02	9.12 ± 12.14	0.377	0.061	0.707
Number of depressive episodes	7.97 ± 9.81	6.50 ± 7.21	1.137	0.171	0.257
Number of manic episodes	3.14 ± 4.45	4.50 ± 7.18	−1.051	0.228	0.296

**Table 2 genes-15-01517-t002:** Number of PRSs divided by category that significantly contributed (VIP > 1) to the prediction of PPD vs. no PPD in the MDD sample.

MDD Sample	Psychiatric PRSs (*n* = 67)	Hormone and Pregnancy PRSs (*n* = 89)	Immune and Inflammation PRSs (*n* = 131)	Sleep and Circadian PRSs (*n* = 54)
PPD	12	10	19	7
No PPD	8	14	28	9
Total	20	24	47	16

**Table 3 genes-15-01517-t003:** Number of PRSs divided by category that significantly contributed (VIP > 1) to the prediction of PPD vs. no PPD in the BD sample.

BD Sample	Psychiatric PRSs (*n* = 67)	Hormone and Pregnancy PRSs (*n* = 89)	Immune and Inflammation PRSs (*n* = 131)	Sleep and Circadian PRSs (*n* = 54)
PPD	3	17	21	9
No PPD	10	15	6	15
Total	13	33	28	24

## Data Availability

The original contributions presented in the study are included in the article/Appendix A, further inquiries can be directed to the corresponding author.

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
