# Peer review of "Disentangling the Genetic Landscape of Peripartum Depression: A Multi-Polygenic Machine Learning Approach on an Italian Sample"

_genes, 2024, doi:10.3390/genes15121517_

Round 1
Reviewer 1 Report
Comments and Suggestions for Authors
This study is using a multi-polygenic score approach and characterized the relationship between genome-wide information and the history of PPD in patients with mood disorders, with the hypothesis that multiple polygenic risk scores (PRSs) could potentially influence the development of PPD. Some points for a major revision can be found below:
Please provide basic statistics in the Results section of the Abstract.
Authors should include in the Introduction or in the Discussion (as it is currently missing) data from quantitative/qualitative studies on birth trauma which is close to what they examine especially regarding the approach/future interventions by healthcare professionals. Please find below some references which can be helpful for writing such as a paragraph (https://doi.org/10.1016/j.wombi.2020.09.016
https://doi.org/10.1016/j.wombi.2020.09.021
doi: 10.4081/hpr.2020.9178).
Was there a sample size estimation for the final sample recruitment? Please justify as this along with the question of whether or not the sample is representative for this population of patients is crucial for the results as well as for the discussion.
Did authors report effect sizes for the statistics?
Why was this specific age limit proposed for the inclusion/exclusion criteria?
How were the diagnoses made?
The PPD and noPPD groups were different regarding their age. Authors should think about rerunning analyses and trying to match the two groups regarding all their demographics so they are comparable.
Author Response
- Please provide basic statistics in the Results section of the Abstract.
We thank the reviewer for highlighting this missing information. As suggested, we added the amount of variance explained by the PRSs in the PLS models for PPD status to the abstract.
“The PLS linear regression in the whole sample defined a model explaining 27.12% of the variance in presence of PPD history, 56.73% of variance among MDD, and 42.96% of variance in BD.” (Page 1, lines 23-24)
- Authors should include in the Introduction or in the Discussion (as it is currently missing) data from quantitative/qualitative studies on birth trauma which is close to what they examine especially regarding the approach/future interventions by healthcare professionals. Please find below some references which can be helpful for writing such as a paragraph (https://doi.org/10.1016/j.wombi.2020.09.016
https://doi.org/10.1016/j.wombi.2020.09.021
doi: 10.4081/hpr.2020.9178).
We thank the reviewer for this suggestion. However, our study focused on the genetic correlates of peripartum depression, and we did not look at environmental factors that are related to PPD. We have also checked the PGS catalog, from where the PRSs of our study were retrieves, and could not identify any PRSs related to birth trauma or birth experience. Therefore, we have decided to not include these suggested citations, but we will take into account this insightful comment considering the effect of birth trauma in future research, when we will study the interactions between biological risk and environmental risk on the development of PPD.
- Was there a sample size estimation for the final sample recruitment? Please justify as this along with the question of whether or not the sample is representative for this population of patients is crucial for the results as well as for the discussion.
We thank the reviewer for the possibility of clarifying this point. Since our sample was retrospective, our sample size was determined by screening the patients that had been recruited in an ongoing larger study of mood disorders (n=435) using our inclusion and exclusion criteria. Also, we found a previous study investigating perinatal mood occurrences that observed a 40-50% prevalence of PPD among major patients affected by major depressive and bipolar disorder (Di Florio et al., 2013). Therefore, we believe our sample is representative of the population of individuals with a history of PPD in patients with mood disorder diagnoses as suggested by the literature. We discussed this point adding this information in the methods section.
“Since our sample was retrospective, our sample size was determined by screening the patients that had been recruited in an ongoing larger study of mood disorders (n=435) using our inclusion and exclusion criteria. We believe our sample is representative of the population of individuals with a history of PPD in patients with mood disorder diagnoses as previous literature has observed a 40-50% prevalence of PPD among major patients affected by major depressive and bipolar disorder [36].” (Page 3-4, lines 147-153)
- Di Florio, A.; Forty, L., Gordon-Smith, K.; Heron, J.; Jones, L.; Craddock, N.; & Jones, I. Perinatal episodes across the mood disorder spectrum. JAMA psychiatry 2013,70, 168-175.
4. Did authors report effect sizes for the statistics?
As suggested by the reviewer, a column for Cohen’s D / Cramer's V was added to table 1 to show the effect sizes of the demographic information (Page 5).
- Why was this specific age limit proposed for the inclusion/exclusion criteria?
These inclusion/exclusion criteria were the ones approved by our local Ethics Committee that were used in the recruitment of the larger sample, that our study sample represents a subset. The minimum age requirement was to ensure the inclusion of only adults in our sample, while patients older than 70 were excluded due to possible complicating factors including comorbidities, cognitive impairment, and difficulty with informed consent.
- How were the diagnoses made?
According to the reviewer’s suggestion, we deepened the information regarding mood disorder diagnosis and PPD status evaluation procedures in the methods section.
“Mood disorder diagnoses were made following a procedure including evaluation and diagnosis based on a psychiatric interview (DSM-IV criteria) by the team responsible for the admission to the ward. The diagnosis was confirmed by a senior psychiatrist specialized in mood disorders using a best-estimate method. This process involved interviews with patients, their family members, and prior healthcare providers, along with a review of available medical records [34].” (Page 3, lines 125-130)
“Peripartum status was documented using patients' medical records based on a clinical interview performed by a psychiatrist to gather patient psychiatric history.” (Page 3, lines 137-139)
- The PPD and noPPD groups were different regarding their age. Authors should think about rerunning analyses and trying to match the two groups regarding all their demographics, so they are comparable.
We thank the reviewer for this input. However, we decided to not include age as predictor within our models specifically because of the statistically significant difference on age between the two groups, since it would have probably masked the contributing effect of genetics. Moreover, to ensure there was no effect of age on PRSs, we correlated age and PRSs and none of the correlations were significant (pFDR>0.05).
Reviewer 2 Report
Comments and Suggestions for Authors
Summary:
The present research studies the genetic risk variables that contribute to peripartum depression (PPD) using partial least squares (PLS) regression in conjunction with a multi-polygenic risk score (PRS) framework. Various PRSs are examined, with a focus on how well they predict PPD in people who already have mood disorders including Major Depressive Disorder (MDD) and bipolar disorder (BD).
Strengths:
1. The application of multi-PRS and PLS regression presents an advanced and nuanced approach for examining genetic factors related to PPD.
2. The manuscript assesses a broad array of PRSs, enhancing the depth of the genetic analysis.
3. The separate evaluation of MDD and BD subgroups provides detailed insights and highlights potential differences in genetic risk profiles.
4. The manuscript effectively interprets and presents the results, making complex genetic findings accessible to readers.
Recommendations:
1. The current figures displaying variable importance or VIP scores come off as repetitive and not particularly informative. consider other visualization methods that provide clearer insights or comparative analysis. Options like heatmaps, boxplots could make the data more engaging and informative.
2. Including a ROC curve to show the sensitivity and specificity would provide more confidence in the model's practical utility.
3. The authors are using PLS regression, but there’s barely any explanation as to why. Why pick this method over others?
4. Cross-validation details are vague. How many folds? What validation strategy?
5. The way that the authors using multiple PRSs is questionable—there's significant overlap that could skew the predictive power.
6. Why focusing on top PRSs with VIP > 1. Why this cutoff?
7. The demographic table (Table 1) feels thin. Adding more clinical characteristics if possible.
8. Reference supplementary tables better. If the authors are going to dump extra data there, at least make it easier for readers to find and understand how it ties into the main analysis.
Conclusion:
The manuscript presents an important contribution to understanding the genetic underpinnings of PPD. The study's overall impact and readability will be strengthened by addressing the recommendations pertaining to statistical transparency, figure quality, and comparative context, which will make it more reliable and available to a larger audience.
Author Response
- The current figures displaying variable importance or VIP scores come off as repetitive and not particularly informative. consider other visualization methods that provide clearer insights or comparative analysis. Options like heatmaps, boxplots could make the data more engaging and informative.
We thank the reviewer for this suggestion. We changed figure 1, 2 and 3 accordingly, condensing all figures in a single picture (Figure 1) and replacing the bar-plots with heatmaps (page 6).
- Including a ROC curve to show the sensitivity and specificity would provide more confidence in the model's practical utility.
We thank the reviewer for this indication. We have included ROC curves for all PLS regression models as can be found in figure 2. We have also updated the methods and results sections to reflect the addition of the ROC curves.
“Finally, sensitivity and specificity values for the first latent variable extracted from each PLS regression model were estimated to calculate the area under the receiver operator curve (AUC) of the model.” (Page 5, lines 219-221)
“The first latent variable of the whole sample PLS model obtained an AUC of 0.823 (Figure 2A).” (Page 6 lines 242-243)
“The first latent variable of the MDD sample PLS model obtained an AUC of 0.948 (Figure 2B).” (Page 7, lines 266-267)
“The first latent variable of the BD sample PLS model obtained an AUC of 0.904 (Figure 2C).” (Page 7, lines 279-280)
- The authors are using PLS regression, but there’s barely any explanation as to why. Why pick this method over others?
We thank the reviewer for highlighting this issue. We have deepened the rationale behind choosing PLS regression for our study.
“We selected PLS regression given its ability to handle high-dimensional predictors, such as the 431 PRSs included in this analysis, which served as predictors. Furthermore, PLS regression is able to condense the information from all the included predictors into latent variables, without removing any of the predictors as other penalized regressions do.” (Page 4, lines 194-198)
- Cross-validation details are vague. How many folds? What validation strategy?
We agree with the reviewer that the information regarding our cross-validation strategy was missing. We performed a standard k-fold cross-validation procedure with 7 folds. Indeed, the numerosity and the imbalance of our sample did not allow us to conduct a hold-out cross validation. Moreover, we did not have any independent cohort to perform an external validation. Details on the cross validation have been added to the manuscript.
“The Nonlinear Iterative Partial Least Squares (NIPALS) algorithm was used to optimize the model by selecting the appropriate number of PLS components to extract (A) via a k-fold cross-validation with 7 folds.” (Page 4-5, lines 201-204)
- The way that the authors using multiple PRSs is questionable—there's significant overlap that could skew the predictive power.
We agree with the reviewer that the overlap among PRSs represents a potential issue within our study, we have therefore added this aspect as a limitation in the manuscript.
“In addition, in our models there are multiple PRSs for the same phenotype that were derived from different source articles using various calculation and imputation methods which may skew the predictive power of the analysis, however we attempted to mitigate this issue by using a PLS regression to combat multicollinearity.” (Page 11, lines 486-489)
- Why focusing on top PRSs with VIP > 1. Why this cutoff?
We selected VIP>1 as this cut-off was suggested by previous literature. Indeed, according to some studies (41-43), a threshold of VIP above 0.8 can be used in for selecting variables in exploratory analyses. These citations have been added to the manuscript.
“[…] with VIP values greater than 1 indicating significant contributions to the model, as suggested by previous literature [41–43].” (Page 5 lines 209-210)
- Cao, K.-A.L.; Rossouw, D.; Robert-Granié, C.; Besse, P. A Sparse PLS for Variable Selection When Integrating Omics Data. Statistical Applications in Genetics and Molecular Biology 2008, 7, doi:10.2202/1544-6115.1390.
- Akarachantachote, N.; Chadcham, S.; Saithanu, K. Cutoff Threshold of Variable Importance in Projection for Variable Selection. International Journal of Pure and Apllied Mathematics 2014, 94, doi:10.12732/ijpam.v94i3.2.
- Chong, I.-G.; Jun, C.-H. Performance of Some Variable Selection Methods When Multicollinearity Is Present. Chemometrics and Intelligent Laboratory Systems 2005, 78, 103–112, doi:10.1016/j.chemolab.2004.12.011.7. The demographic table (Table 1) feels thin. Adding more clinical characteristics if possible.
We thank the reviewer for this suggestion. We therefore enlarged the set of demographic characteristics of our sample. Specifically, BMI, smoking history, alcohol history, sex of children, menopause status, and current hormone replacement therapy variables have been added to the demographic table (table 1).
8. Reference supplementary tables better. If the authors are going to dump extra data there, at least make it easier for readers to find and understand how it ties into the main analysis.
We thank the reviewer for the opportunity of ameliorating the reference to supplementary tables. We updated the manuscript accordingly.
“For reference, a full list of all the PRSs included in our study and their corresponding source articles including the information regarding the original GWAS studies and their Polygenic Score Catalog codes is provided in Supplementary Table 1.” (Page 4, lines 184-187).
“The top 20 contributing PRSs can be found in Figure 1A and the full results showing the ranking and weight of the 106 PRSs with a VIP greater than 1, and thus are significant contributors to the PPD prediction model, can be found in Supplementary Table 2.” (Page 6, lines 240-242)
“The top 20 contributing PRSs can be found in Figure 1B and the full results showing the ranking and weight of the 107 PRSs with a VIP greater than 1, and thus are significant contributors to the PPD prediction model, can be found in Supplementary Table 3.” (Page7, lines 263-266)
“The top 20 contributing PRSs can be found in Figure 1C and the full results showing the ranking and weight of the 98 PRSs with a VIP greater than 1, and thus are significant contributors to the PPD prediction model, can be found in Supplementary Table 4.” (Page 7, lines 276-279)
Reviewer 3 Report
Comments and Suggestions for Authors
Comments:
1. Please list all abbreviations.
2. If all patients were from Italy, please add ".... scores in Italy" in title.
3. Table 1: Please add more patients' info, such as BMI/BRI, TSH/T4/T3 level, any GI issues, smoking and alcohol history, current hormone replacement therapy, etc.
4. Please explain why there are few or no young patients? Please list the age rang in Table 1.
5. Please add the gender of kids with the number of pregnancies.
6. Please explain and discuss why age and education are the only 2 factors with significant difference in Table 1.
Author Response
- Please list all abbreviations.
According to the journal guidelines, a list of abbreviations is not required in the manuscript, but all of them are defined at first mention in the abstract and main text. Here below we provide a list of the abbreviations here for the reviewer’s reference.
PPD: Peripartum depression
PRSs: Polygenic risk scores
MDD: Major depressive disorder
BD: Bipolar disorder
SNPs: Single nucleotide polymorphisms
GWAS: Genome wide association studies
PLS: Partial least squares
DSM: Diagnostic and Statistical Manual of Mental Disorders
QC: Quality control
MAF: Minor allele frequency
PCA: Principal component analysis
NIPALS: Nonlinear Iterative Partial Least Squares
W: Weights
VIP: Variable importance in projection
RCTs: Randomized controlled trials
CCL3: Chemokine (C-C motif) ligand 3
CRP: C-reactive protein
IL: Interleukin
PGF: Placenta growth factor
SHBG: Sex hormone binding globulin
- If all patients were from Italy, please add ".... scores in Italy" in title.
We thank the reviewer for this suggestion, and we have changed the title of the manuscript accordingly.
“Disentangling the Genetic Landscape of Peripartum Depression: A Multi-Polygenic Machine Learning Approach on an Italian Sample” (page 1, lines 2-4).
- Table 1: Please add more patients' info, such as BMI/BRI, TSH/T4/T3 level, any GI issues, smoking and alcohol history, current hormone replacement therapy, etc.
We thank the reviewer for this suggestion, which is also in agreement with another reviewer’s suggestion of adding more clinical information of our sample. We therefore have added BMI, smoking history, alcohol history, sex of children, menopause status, and current hormone replacement therapy in the demographics table. Unfortunately, TSH/T4/T3 levels and presence of GI issues information were not available. Please see the updated demographics table on page 5.
- Please explain why there are few or no young patients? Please list the age range in Table 1.
Our sample was retrieved from a larger cohort, whose inclusion and exclusion criteria were approved by the local Ethics Committee. Indeed, the minimum age requirement of 18 years old was to ensure the inclusion of only adults in our sample, while patients older than 70 were excluded due to possible complicating factors including comorbidities, cognitive impairment, and difficulty with informed consent. The age range of participants has been included in the participants subsection of the methods section.
“We specifically analyzed 178 parous women aged between 28 and 70 diagnosed with MDD or BD, 62 of whom had a history of PPD, while 116 had no such history.” (Page 3, lines 132-133)
- Please add the gender of kids with the number of pregnancies.
We thank the reviewer for this suggestion. Sex of the children has been added as a variable in the demographics table 1. Please see the updated demographics table on page 5.
- Please explain and discuss why age and education are the only 2 factors with significant difference in Table 1.
We thank the reviewer for the possibility of further explaining these findings. Age was the only factor significantly different between the two groups. PPD women may be significantly younger as they had a depressive episode during their reproductive window, while women who do not have a PPD history may be older as they may not have developed depressive symptomatology until later in life. As far as education is concerned, in our sample while the PPD women seem to have slightly more years of education, this difference is not statistically different. We have added a sentence discussing the difference in age to the results section.
“Age was the only factor significantly different between the two groups. PPD women may be significantly younger as they had a depressive episode during their reproductive window, while women who do not have a PPD history may be older as they may not have developed depressive symptomatology until later in life.” (Page 5, lines 223-227).
Round 2
Reviewer 3 Report
Comments and Suggestions for Authors
No more comments